# An AI Agent for Immune Receptor Fingerprint-Based Diagnosis of Infection of Unknown Origin

## Abstract

When routine tests fail to find a pathogen, diagnosing infections of unknown origin stalls. We instead read the patient's immune response for AI-readable clues. We formalize a new machine learning task: inferring plausible epitopes directly from immune-receptor repertoires and localizing their pathogen sources. To address this problem, we introduce a Transformer-based multi-sequence novel representation-learning model that jointly models T-cell receptors, human leukocyte antigen , and antigenic peptides, and we pretrain it across six tasks; the model achieves best or second-best performance across all six tasks against strong baselines. Building on this, we develop an end-to-end, clinically oriented agent that operates in a perceive–plan–act loop, orchestrating epitope generation, HLA-personalized filtering, consistency checks, and retrieval, with clinician-in-the-loop threshold adaptation; when evidence conflicts, it performs calibrated abstention and logs an interpretable decision trace. End-to-end on clinical-style repertoires with diagnostic report generation, the agent outperforms discriminative-pairing and direct-retrieval baselines. Upon publication, we will release all code, models, and pathogen indices under a research license, together with de-identified evaluation data.

## 1 Introduction

Clinically, a large number of patients present with suspected infection but no identified pathogen, leading to delayed or inappropriate therapy, prolonged hospital stays, repeat imaging and invasive procedures, higher costs, avoidable morbidity, and the unnecessary use of broad-spectrum antimicrobials that fuel resistance (de Jonge et al., 2022; Tong et al., 2021). As emphasized in clinical guidance (e.g., IDSA and ESCMID recommendations), when cultures are negative and multiplex PCR panels have limited coverage, diagnosing infections of unknown origin (IUO) is especially difficult and clinicians are left without timely, actionable leads (Lawandi et al., 2022). Meanwhile, the adaptive immune system encodes a history of antigen exposure within the patient's human leukocyte antigen (HLA) context, with T-cell receptor (TCR) repertoires effectively "recording" that history (Zaslavsky et al., 2025). This host-side signal is scalable to collect, tightly coupled to antigen exposure, and naturally suited to machine learning and AI models. However, most existing approaches cast TCR–peptide recognition as a narrow binary classification problem or rely solely on direct sequence retrieval, limiting generalization, calibration, and the ability to generate novel antigen hypotheses.

**This work introduces a new computational problem**: using only a patient's TCR repertoire and HLA type—without any pathogen reads—we aim to infer plausible epitopes and localize their pathogen sources. We cast this as a representation-learning and generative modeling task: generate plausible peptide epitopes from immune-receptor data and map them to concrete pathogens via a generate–then–retrieve pipeline. This formulation imposes two core requirements: (i) a model that, under limited supervision, learns allele specificity and cross-reactivity while producing calibrated uncertainty; and (ii) a retrieval agent that can break out of a fixed database when evidence is ambiguous and elastically expand its search index on demand, without retraining.

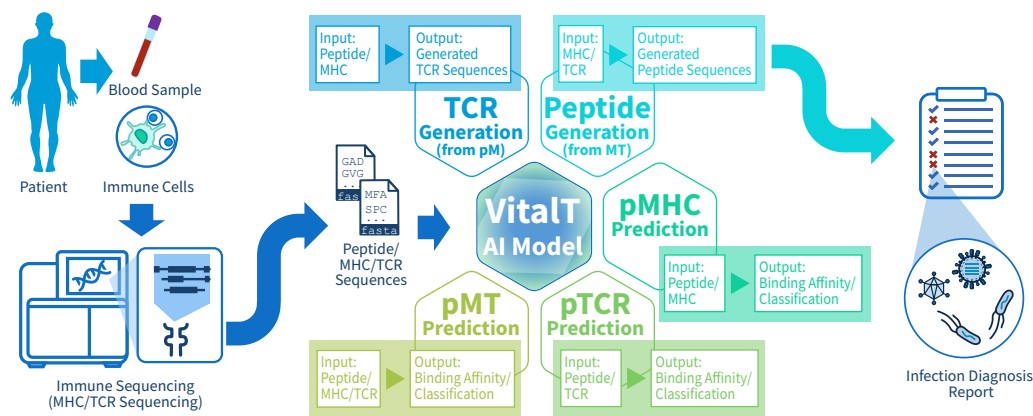

Figure 1: An agent for immune receptor fingerprint-based diagnosis of infections of unknown origin (IUO). It follows a generate–then–retrieve paradigm: hypothesize epitopes from the host signal, link them to an expandable pathogen proteome index, and aggregate evidence to rank likely pathogens with optional abstention under uncertainty.

To meet these demands, **this work introduce a new Transformer-based multi-sequence representation-learning model** that jointly encodes TCRs, HLA, and peptides. We pretrain it with six complementary objectives that reflect how evidence is used in downstream decisions: three classification tasks , one discriminative objective that shapes predictive uncertainty for selective prediction and abstention, and two conditional generation tasks that synthesize sequences in both directions. This design encourages allele-aware semantics and cross-reactivity modeling while producing uncertainty estimates that are stable enough to drive an agent's control logic.

Building on this foundation, **this work develop an end-to-end, clinically oriented agent** that operates in a perceive–plan–act loop. The agent resamples peptides when uncertainty is high, applies HLA-personalized filtering, performs cross-clonotype consistency checks, and supports abstain/escalate with adaptive thresholds under a clinician-in-the-loop. It then conducts generate–then–retrieve matching over a large pathogen proteome index. Crucially, the retrieval corpus is dynamically expandable without retraining: the agent can widen its search radius by streaming additional proteomes into the index on demand, enabling effectively unbounded search when initial evidence is inconclusive.

We evaluate on clinical-style repertoires curated from public resources (Goncharov et al., 2022). We compare against strong sequence baselines (ANN, CNN, LSTM, VAE) and a pretrained protein language model, and assess both classification and generation metrics. Across tasks, our method achieves best or second-best performance on accuracy, AUROC and F1 Score, as well as on token- and sequence-level generation metrics (token accuracy, sequence accuracy, perfect ratio). End to end, the agent improves peptide reconstruction and pathogen localization, remains robust under sparse/noisy repertoires, and yields interpretable decision traces with calibrated uncertainty. We also include an agent-generated clinical report for demonstration.

**Contributions.** We summarize our contributions in three parts:

- **New problem.** We formalize host-only IUO diagnosis as generative epitope inference with pathogen localization from immune-receptor data, shifting focus from pathogen reads to a representation learning and generative perspective on the host signal.

- **New model.** We present a Transformer-based multi-sequence model with six objectives—three classification (PT, PM, PMT), one discriminative calibration objective for uncertainty, and two conditional generation objectives for peptides and TCRs—that advances both *classification* (accuracy, AUROC, F1, MCC) and *generation* (token accuracy, sequence accuracy, perfect ratio) metrics over strong baselines.

- **New agent.** We build an uncertainty-aware clinical agent that couples generate–then–retrieve with HLA-personalized filtering, consistency checks, abstention, and clinician-in-the-loop thresholding, while supporting *on-demand, unbounded expansion* of the retrieval index without retraining. We release code, models, and pathogen indices under a research license with a reproducible evaluation suite and de-identified data upon publication.

## 2 RELATED WORK

**Host-based diagnostics for infectious disease.** Host-side signals have long been explored for diagnosing infection and stratifying severity, from blood transcriptomic signatures to immune-receptor profiling (Cairns et al., 2010; Zaslavsky et al., 2025). These approaches reduce dependence on pathogen reads and can be sampled at scale, but most either require bespoke assay panels or offer limited pathogen localization. Our work remains host-only yet aims to produce actionable, pathogen-linked hypotheses by coupling immune-receptor modeling with generate–then–retrieve.

**Immune-repertoire modeling and TCR specificity.** Early repertoire methods relied on motif discovery, distance-based clustering, or k-mer statistics to group clonotypes with shared specificity (Mayer-Blackwell et al., 2022). Subsequent deep models framed TCR–peptide recognition as supervised classification with CNN/LSTM/attention encoders (Sidhom et al., 2021; Montemurro et al., 2021), while triadic formulations explicitly incorporate MHC context (PMT) (Lu et al., 2021; Ma et al., 2025). These works advance discriminative accuracy but typically (i) operate in closed-world datasets, (ii) provide limited uncertainty handling for selective prediction, and (iii) do not generate new antigenic hypotheses from host data. We instead pair multi-task representation learning with conditional sequence generation in both directions enabling hypothesize-first reasoning.

**Protein language models and sequence generation.** Large protein LMs such as ESM and Prot-BERT learn transferable representations for diverse downstream tasks, including structure-aware scoring and sequence design (Ho et al., 2026; Hao & Fan, 2024). In immunology, they have been used to embed peptides and sometimes TCRs, predominantly within discriminative pipelines. Our model departs in two ways: (i) we pretrain jointly on TCR, peptide, and HLA with six objectives aligned to the agent's decisions (three classifications + uncertainty shaping + two conditional generations), and (ii) we exploit generation explicitly to propose candidate epitopes before retrieval, improving open-world generalization.

**Structural modeling of TCR–pMHC interactions.** Structure- or docking-based methods can estimate biophysical compatibility and provide mechanistic insight (Yin et al., 2023; Wu et al., 2025). However, they are computationally intensive, sensitive to modeling choices, and difficult to scale for cohort-level repertoire data. Our approach is sequence-based and agentic, trading fine-grained structural accuracy for throughput and uncertainty-aware decision-making, and can later incorporate structural checks as optional tools.

**Retrieval over proteomes and retrieval-augmented generation.** Classical sequence search engines (e.g., BLAST, JackHMMER and MMseqs2) enable fast similarity queries across large proteomes (Johnson et al., 2010; Kallenborn et al., 2025). In NLP, retrieval-augmented generation (RAG) improves factuality and open-domain generalization by coupling generators with expandable indices. Our agent adopts a generate–then–retrieve paradigm for biology: we first propose epitopes from host data and then match them against a pathogen proteome index that can be dynamically expanded without retraining, enabling effectively unbounded search when evidence is ambiguous.

## 3 PRELIMINARIES

**Biological and clinical background.** Cytotoxic T cells recognize short peptides presented by HLA molecules on the surface of host cells. These peptide–HLA complexes (pHLA) are sampled by T-cell receptors (TCRs); clonotypes specific to the ongoing infection expand and leave measurable footprints in the host immune repertoire. High-throughput repertoire sequencing provides $(R, H)$ pairs per individual, where $R$ summarizes the observed receptors and $H$ the HLA type. Our

host-only diagnostic setting seeks to convert these repertoire-level signals into actionable, pathogen-linked hypotheses. The formulation primarily targets HLA class I and can be naturally extended to class II.

**Problem setup and notation.** Let $\mathcal{A}$ be the 20-letter amino-acid alphabet and $\mathcal{A}^*$ the space of finite sequences. We denote: - $\mathcal{T} \subseteq \mathcal{A}^*$: the space of TCR sequences, $t \in \mathcal{T}$; - $\mathcal{E} \subseteq \mathcal{A}^*$: the space of epitope peptides, $e \in \mathcal{E}$; - $\mathcal{H}$: the discrete set of HLA alleles, with an individual's HLA type $H \subseteq \mathcal{H}$; - $\mathcal{G}$: an index set of candidate pathogen identities; for $g \in \mathcal{G}$, $\mathcal{P}(g) \subseteq \mathcal{A}^*$ collects the proteins attributed to $g$.

A clinical instance is $(R, H)$, where $R$ is a finite multiset (empirical measure) of receptors,

$$R = \left\{ (t_i, w_i) \right\}_{i=1}^m, \quad t_i \in \mathcal{T}, \ w_i \geq 0, \ \sum_i w_i = 1,$$

equivalently $\mu_R = \sum_i w_i \, \delta_{t_i}$ over $\mathcal{T}$.

**Latent targets and objectives.** Assume a latent set of immunodominant epitopes $Z^\star = \{e_j^\star\}_{j=1}^M \subseteq \mathcal{E}$ and a latent pathogen label $g^\star \in \mathcal{G}$ such that $e_j^\star \in \mathcal{P}(g^\star)$. We observe only $(R, H)$. The agent pursues two objectives: (i) Generation: learn a conditional distribution $q_\theta(e \mid R, H)$ over $\mathcal{E}$ and propose $\hat{Z} = \{(e_k, p_k)\}_{k=1}^K$ with $p_k = q_\theta(e_k \mid R, H)$; (ii) Localization: score/rank $\mathcal{G}$ based on how proposed peptides could arise from each candidate.

**Agent view.** We adopt an agentic abstraction with two decision primitives: (1) a conditional generator producing peptides as character-level sequences over $\mathcal{A}$; (2) a retrieval operator that links proposed peptides to a proteome index. A separate decision head can abstain when confidence is low; implementation is detailed in Methods.

**Simple fuzzy retrieval via string identity on a sliding-window peptide library.** For any protein set $\mathcal{P}(g)$ and length $L \in \mathbb{N}$, define the sliding-window library

$$\mathsf{Win}(g, L) = \{x \in \mathcal{A}^L : \ x \text{ is a contiguous substring of some } p \in \mathcal{P}(g)\}.$$

For equal-length strings $e, x \in \mathcal{A}^L$, define the position-wise identity

$$\mathrm{sim}(e, x) = \frac{1}{L} \sum_{j=1}^{L} \mathbb{1}\{e_j = x_j\} \in [0, 1].$$

The retrieval score for a peptide and an index $g$ is the best-window match:

$$\mathcal{R}(e, g) = \max_{x \in \mathsf{Win}(g, |e|)} \mathrm{sim}(e, x).$$

Given candidates $\{(e_k, p_k)\}_{k=1}^K$, we aggregate to a pathogen score

$$S_\theta(g \mid R, H) \ = \ \sum_{k=1}^{K} p_k \, \mathcal{R}(e_k, g),$$

and return the Top-$k$ set $\hat{\mathcal{G}}_k(R, H)$ accordingly. Any thresholds or engineering choices (e.g., indexing, pruning) are deferred to Methods.

**Supervision and discriminative sub-task.** When labels are available, we consider tuples $(t, e, h, y)$ with $y \in \{0, 1\}$ indicating interaction feasibility under allele context $h$, which support auxiliary discriminative objectives $r_\theta(t, e, h) \approx \mathbb{P}(y{=}1 \mid t, e, h)$. Concrete loss design and multi-task coupling are in Methods.

**Uncertainty and selective outputs.** The agent reports confidences for candidates ($u_e$) and for pathogen scores ($u_g$), and may abstain ($\perp$) below a threshold. Formal calibration criteria and evaluation live in Methods.

**Evaluation endpoints.** - Generation fidelity: when references $Z^\star$ exist, sequence- and character-level correctness for $\hat{Z}$; - Localization success: $\text{Succ}_k = \mathbb{1}\{g^\star \in \hat{\mathcal{G}}_k(R, H)\}$; - Selective metrics: coverage–risk trade-offs under varying thresholds (details in Methods).

## 4 METHODS

We instantiate the generate–then–retrieve agent introduced in Preliminaries with a shared Transformer encoder for multi-task discrimination, two lightweight causal decoders for conditional sequence generation, and a minimal fuzzy retrieval module that links generated peptides to a proteome index. The method operates on serialized amino-acid sequences augmented with a few special tokens (e.g., separators and end-of-sequence). Unless otherwise stated, we focus on HLA class I and discuss extensions in the Appendix.

### 4.1 SEQUENCE SERIALIZATION AND SHARED ENCODER

All inputs are serialized as token sequences over an augmented alphabet and fed to a shared encoder. Given an input token matrix $X \in \{0, \dots, V-1\}^{B \times L}$ (batch size $B$, length $L$), we apply token embeddings $E \in \mathbb{R}^{V \times d}$ and positional embeddings $P \in \mathbb{R}^{L \times d}$, and obtain $Z^{(0)} = \text{LN}(\text{Drop}(XE + P))$. A stack of $N$ standard Transformer layers (multi-head self-attention + MLP with GELU and residual pre-LayerNorm) yields contextual representations $Z^{(N)} \in \mathbb{R}^{B \times L \times d}$. For any downstream head, we compute a masked mean-pooled vector

$$\bar{z} = \frac{\sum_{i=1}^{L} m_i Z_{:,i,:}^{(N)}}{\sum_{i=1}^{L} m_i}, \quad m_i \in \{0, 1\},$$

followed by a small projection $\tilde{z} = \tanh(W_p \bar{z})$.

### 4.2 MULTI-TASK BINARY CLASSIFIERS WITH AUXILIARY HEADS

We use three binary classifiers that share the encoder: (i) PT: peptide–TCR context; (ii) PMT: peptide–MHC–TCR context; (iii) PM: peptide–MHC context. Each classifier maps $\tilde{z}$ to logits $o \in \mathbb{R}^2$ via a linear head. Two auxiliary heads are exposed: (a) a scalar confidence proxy $c_\theta = \sigma(w_c^\top \tilde{z})$; and (b) a plausibility discriminator $D_\phi(\tilde{z}) \in (0, 1)$, implemented as a small MLP. The stage-1 loss is

$$\mathcal{L}_{\text{stage-1}} = \sum_{t \in \{\text{PT,PMT,PM}\}} \text{CE}(o^{(t)}, y) + \lambda_{\text{disc}} \text{BCE}(D_\phi(\tilde{z}), y),$$

with $\lambda_{\text{disc}} = 0.1$ by default. The confidence proxy is not directly supervised; calibration and selective-use protocols are evaluated separately.

### 4.3 CAUSAL DECODERS FOR CONDITIONAL GENERATION

We employ two GPT-style causal decoders with independent token/positional embeddings and the same Transformer building block as the encoder, but under a standard causal mask. These decoders are designed to model conditional generation in both directions:

**TCR Generator:** Models $q_\theta^{\text{TCR}}(t \mid e, h)$ from a serialized context such as "`peptide <SEP> MHC <SEP>`".

**Peptide Generator:** Models $q_\theta^{\text{pep}}(e \mid t, h)$ from a context such as "`MHC <SEP> TCR <SEP>`".

Both decoders optimize next-token cross-entropy with a single right-shift and ignore padding tokens during training. At inference, we sample with temperature and top-$k$ filtering, stopping at `<EOS>` tokens and skipping special tokens in the final readout. These decoders instantiate the conditional generator $q_\theta(\cdot \mid R, H)$ used by the agent for generating biologically plausible sequences.

**Training schedule** We train in three stages with targeted freezing: (1) Stage 1 (discrimination): train the three classifiers and the discriminator; freeze both generators. (2) Stage 2 (TCR generation): train the TCR decoder; freeze classifiers, discriminator, and peptide decoder. (3) Stage 3 (peptide generation): train the peptide decoder; freeze all others.

**Inference APIs** The agent exposes: (a) binding feasibility prediction for PT/PMT/PM by running the encoder and the corresponding classifier (with optional use of $c_\theta$ as a proxy confidence); (b) conditional generation via either decoder given a serialized context; and (c) generate–then–retrieve diagnosis, where generated peptide candidates are passed to the retrieval module below.

## 4.4 FUZZY RETRIEVAL OVER A SLIDING-WINDOW PROTEOME INDEX

For each candidate pathogen $g$, we maintain a proteome index $\mathcal{P}(g)$ and define a sliding-window library of contiguous substrings with the same length as a query peptide $e$:

$$\mathsf{Win}(g, |e|) = \{x \in \mathcal{A}^{|e|} : x \text{ is a contiguous substring of some } p \in \mathcal{P}(g)\}.$$

We use the simplest position-wise identity as similarity for equal-length strings,

$$\mathrm{sim}(e, x) = \frac{1}{|e|} \sum_{j=1}^{|e|} \mathbb{1}\{e_j = x_j\} \in [0, 1],$$

and define the retrieval operator

$$\mathcal{R}(e, g) = \max_{x \in \mathsf{Win}(g, |e|)} \mathrm{sim}(e, x).$$

Given a set of $K$ generated candidates $\hat{Z} = \{e_k\}_{k=1}^K$, we aggregate to a pathogen score by a simple unweighted average

$$S_\theta(g \mid R, H) = \frac{1}{K} \sum_{k=1}^K \mathcal{R}(e_k, g),$$

and return a Top-$k$ list over $g \in \mathcal{G}$. A sample can also be flagged as positive for a specific $g$ if $\max_k \mathcal{R}(e_k, g) \geq \tau$ for a user-chosen threshold $\tau \in [0, 1]$. Indexing, I/O, and batching strategies are standard and omitted here.

## 5 EXPERIMENTS

### 5.1 EXPERIMENTAL SETUP

**Datasets and splits.** Public repositories such as IEDB, VDJdb, and McPAS-TCR have standardized evaluation but can induce closed-world biases (Vita et al., 2025; Goncharov et al., 2022; Yue et al., 2025). We curate out-of-training-set, clinical-style repertoires and assess both classification (accuracy, AUROC, F1, MCC) and generation (token/sequence accuracy, perfect-ratio) metrics, with agent-level endpoints aligning evaluation with open-world use.

**Tasks and metrics.** For PT/PM/PMT classification we report Accuracy, F1, ROC-AUC, and PR-AUC. For generation we report token-level accuracy and perplexity; when references are available, we also report sequence-level match.

### 5.2 STAGE-WISE PERFORMANCE ON GENERATION AND DISCRIMINATION

Figure 2 summarizes stage-wise metrics that jointly assess generative quality and pairwise discrimination across three training stages. Stage 1 evaluates the classifier/discriminator stack, while Stages 2 and 3 focus on conditional sequence generation for TCRs and peptide epitopes, respectively. For generation, we report Token Accuracy, Sequence Accuracy, and Perfect-Sequence Ratio for both the TCR and peptide decoders. For discrimination, we report ROC AUC, Accuracy, F1, Balanced Accuracy, and MCC under three pairing settings (pM, pT, pMT).

Across stages, the trajectories indicate that improvements in generative fidelity are accompanied by stronger pairwise discrimination. In particular, token- and sequence-level agreement increase alongside gains in imbalance-robust metrics (Balanced Accuracy, MCC), suggesting that enhanced conditional generation translates into more reliable downstream matching and classification. All metrics are shown on a 0–1 scale with higher values being better.

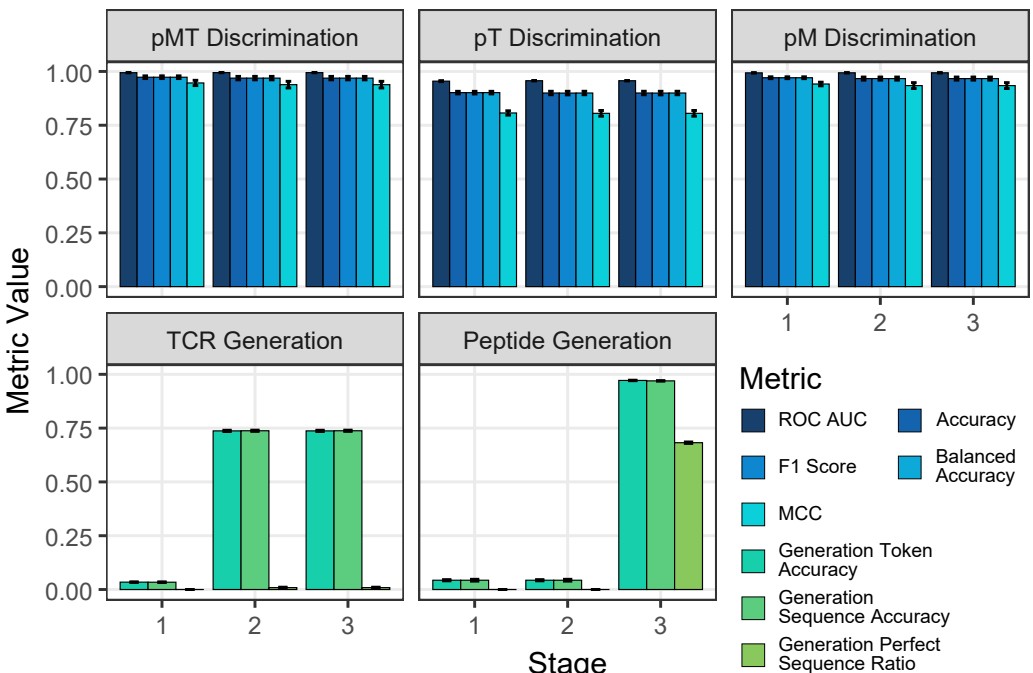

Figure 2: Stage-wise metrics for generation and discrimination. Panels group (i) TCR and peptide generation (Token Accuracy, Sequence Accuracy, Perfect-Sequence Ratio) and (ii) pairwise discrimination for pM, pT, and pMT (ROC AUC, Accuracy, F1, Balanced Accuracy, MCC). The x-axis denotes training Stage (1–3); the y-axis reports metric values in $[0, 1]$, higher is better. Points/lines are means aggregated over five cross-validation folds and multiple seeds (total $>50k$ examples); error bars show 95% confidence intervals, with standard errors shown where applicable.

Table 1: Baseline comparison on classification and generation tasks. Performance metrics (mean $\pm$ 95% CI) highlighting competitive performance and unique capabilities of our approach.

| Model | Classification AUC | | | Generation Quality | |
| | pMT AUC | pM AUC | pT AUC | Peptide Token Acc. | TCR Token Acc. |
|---|---|---|---|---|---|
| ANN | $0.975 \pm 0.002$ | $0.972 \pm 0.002$ | $0.965 \pm 0.003$ | $0.021 \pm 0.001$ | $0.068 \pm 0.002$ |
| CNN | $0.982 \pm 0.002$ | $0.976 \pm 0.002$ | $0.968 \pm 0.003$ | $0.028 \pm 0.002$ | $0.074 \pm 0.003$ |
| LSTM | $0.978 \pm 0.002$ | $0.974 \pm 0.002$ | $0.962 \pm 0.004$ | $0.035 \pm 0.046$ | $0.082 \pm 0.009$ |
| VAE | $0.973 \pm 0.003$ | $0.969 \pm 0.003$ | $0.955 \pm 0.005$ | $0.032 \pm 0.010$ | $0.089 \pm 0.014$ |
| ESM-2 | $0.979 \pm 0.002$ | $0.977 \pm 0.002$ | $0.963 \pm 0.004$ | $0.025 \pm 0.005$ | $0.076 \pm 0.002$ |
| **VitalT (This Work)** | $\mathbf{0.996 \pm 0.002}$ | $\mathbf{0.996 \pm 0.001}$ | $\mathbf{0.984 \pm 0.001}$ | $\mathbf{0.045 \pm 0.001}$ | $\mathbf{0.110 \pm 0.008}$ |

## 5.3 BASELINE COMPARISON

We benchmark our model against ANN, CNN, LSTM, VAE, and ESM-2 on three classification datasets and two generation tasks; results are summarized in Table 1. Our approach attains the highest classification AUC across all settings, with 0.996 on pMT, 0.996 on pM, and 0.984 on pT. The next best scores are 0.982 on pMT (CNN), 0.977 on pM (ESM-2), and 0.968 on pT (CNN), giving absolute gains of 0.014, 0.019, and 0.016. The reported 95% confidence intervals do not overlap between our model and the strongest baselines on classification, indicating consistent improvements under the same protocol. On sequence generation, our model also yields the best token accuracies, 0.110 for TCR and 0.045 for peptides. Relative to the strongest baselines, this corresponds to improvements of 0.021 over VAE for TCR and 0.010 over LSTM for peptides, with tighter confidence intervals. Overall, the method combines state-of-the-art discrimination with strong sequence modeling, and we therefore use it as the default system in subsequent experiments.

## 5.4 ABLATION STUDY

Table 2: Ablation study for classification with component configuration (left) and discrimination metrics (mean ± 95% CI) on pT, pMT, and pM datasets. Components: MT = multi-task learning; Gen = sequence generation; PE = positional encoding; LN = layer normalization; Disc = adversarial discriminator.

| Model Variant | MT | Gen | PE | LN | Disc | Size | pT | | pMT | | pM | |
|---|---|---|---|---|---|---|---|---|---|---|---|---|
| | | | | | | | F1 | AUC | F1 | AUC | F1 | AUC |
| **Baseline (Full Model)** | ✓ | ✓ | ✓ | ✓ | ✓ | M | 0.9533 ± 0.0018 | 0.9835 ± 0.0004 | 0.9813 ± 0.0020 | 0.9964 ± 0.0009 | 0.9797 ± 0.0009 | 0.9961 ± 0.0011 |
| w/o Positional Encoding | ✓ | ✓ | ✗ | ✓ | ✓ | M | 0.9231 ± 0.0009 | 0.9673 ± 0.0006 | 0.9705 ± 0.0012 | 0.9936 ± 0.0006 | 0.9763 ± 0.0005 | 0.9957 ± 0.0006 |
| w/o Layer Normalization | ✓ | ✓ | ✓ | ✗ | ✓ | M | 0.9175 ± 0.0103 | 0.9655 ± 0.0068 | 0.9789 ± 0.0023 | 0.9953 ± 0.0007 | 0.9785 ± 0.0025 | 0.9950 ± 0.0005 |
| w/o Discriminator | ✓ | ✓ | ✓ | ✓ | ✗ | M | 0.9532 ± 0.0049 | 0.9834 ± 0.0015 | 0.9821 ± 0.0012 | 0.9969 ± 0.0001 | 0.9797 ± 0.0008 | 0.9966 ± 0.0001 |
| w/o Generation | ✓ | ✗ | ✓ | ✓ | ✗ | M | 0.9536 ± 0.0018 | 0.9827 ± 0.0006 | 0.9810 ± 0.0011 | 0.9966 ± 0.0003 | 0.9798 ± 0.0009 | 0.9965 ± 0.0003 |
| Single Task: pT only | ✗ | ✗ | ✓ | ✓ | ✗ | M | 0.9462 ± 0.0027 | 0.9797 ± 0.0014 | — | — | — | — |
| Single Task: pMT only | ✗ | ✗ | ✓ | ✓ | ✗ | M | — | — | 0.9807 ± 0.0004 | 0.9965 ± 0.0001 | — | — |
| Small Model (S) | ✓ | ✓ | ✓ | ✓ | ✓ | S | 0.9324 ± 0.0029 | 0.9726 ± 0.0027 | 0.9796 ± 0.0003 | 0.9963 ± 0.0003 | 0.9782 ± 0.0009 | 0.9960 ± 0.0004 |
| Large Model (L) | ✓ | ✓ | ✓ | ✓ | ✓ | L | 0.9448 ± 0.0030 | 0.9791 ± 0.0011 | 0.9804 ± 0.0019 | 0.9963 ± 0.0001 | 0.9793 ± 0.0013 | 0.9963 ± 0.0002 |

We ablate one component at a time under a fixed training protocol and report means with 95% confidence intervals (Tables 2 and 3). Two components are critical: removing positional encoding or layer normalization yields the largest and most consistent drops, especially on pT. The pT F1 decreases from 0.9533 to 0.9231 and 0.9175, and the pT AUC decreases from 0.9835 to 0.9673 and 0.9655, corresponding to absolute changes of 0.030 and 0.036 for F1 and 0.016 and 0.018 for AUC. Declines on pMT and pM follow the same pattern but are smaller. In contrast, removing the discriminator or disabling generation leaves classification essentially unchanged, with pT F1 near 0.953 and AUC near 0.983, indicating that the encoder with positional encoding and layer normalization drives discrimination performance. Capacity ablations favor the medium configuration: the small model underfits (pT F1 0.9324, AUC 0.9726), while the large model does not improve pT and yields only marginal differences on pMT and pM. For sequence generation (Table 3), removing positional encoding or layer normalization reduces TCR token accuracy from 0.110 to about 0.101 and 0.102 and slightly lowers peptide token accuracy, while removing the discriminator has limited effect. We therefore retain positional encoding and layer normalization and use the medium model in subsequent experiments.

Table 3: Ablation study for sequence generation with component configuration (left) and generation metrics (mean ± 95% CI) on TCR and peptide generation tasks. Components: MT = multi-task learning; Gen = sequence generation; PE = positional encoding; LN = layer normalization; Disc = adversarial discriminator.

| Model Variant | MT | Gen | PE | LN | Disc | Size | TCR Generation | | Peptide Generation | |
|---|---|---|---|---|---|---|---|---|---|---|
| | | | | | | | Token Acc. | Loss | Token Acc. | Loss |
| **Baseline (Full Model)** | ✓ | ✓ | ✓ | ✓ | ✓ | M | 0.110 ± 0.008 | 81.38 ± 9.85 | 0.045 ± 0.001 | 82.67 ± 2.86 |
| w/o Positional Encoding | ✓ | ✓ | ✗ | ✓ | ✓ | M | 0.101 ± 0.000 | 89.71 ± 9.63 | 0.044 ± 0.002 | 89.15 ± 2.58 |
| w/o Layer Normalization | ✓ | ✓ | ✓ | ✗ | ✓ | M | 0.102 ± 0.003 | 14.43 ± 0.83 | 0.045 ± 0.001 | 16.40 ± 0.53 |
| w/o Discriminator | ✓ | ✓ | ✓ | ✓ | ✗ | M | 0.108 ± 0.012 | 72.18 ± 13.75 | 0.043 ± 0.002 | 80.46 ± 12.34 |
| Small Model (S) | ✓ | ✓ | ✓ | ✓ | ✓ | S | 0.127 ± 0.027 | 26.72 ± 1.85 | 0.046 ± 0.002 | 28.63 ± 1.72 |
| Large Model (L) | ✓ | ✓ | ✓ | ✓ | ✓ | L | 0.101 ± 0.001 | 358.02 ± 23.32 | 0.048 ± 0.006 | 413.05 ± 15.95 |

## 5.5 CASE DEMONSTRATION

We executed the full pipeline on a held-out batch of 60 TCR samples. Inference used a similarity threshold of 0.65, temperature 0.8, top-k sampling with k=5, and up to 20 candidate peptides per query against an index of ten pathogens. The run returned five positives (8.3%) spanning three taxa: Salmonella enterica, Mycobacterium tuberculosis, and human immunodeficiency virus 1 (HIV-1). Mean similarities among the positives for these taxa were 0.667, 0.697, and 0.667, respectively. Table 4 lists representative TCR-epitope pairings for the detected pathogens; the complete per-sample report is provided in Appendix A.

Table 4: Detailed TCR-epitope interactions for detected pathogens. We report TCR sequences, generated epitopes, reference epitopes, and similarity scores for positive predictions.

| Pathogen | TCR Sequence | Generated Epitope | Reference Epitope | Similarity |
|---|---|---|---|---|
| Salmonella enterica | CASSLEASSYNSPLHF | FAFLDPADLAQ | FSLPAQDLVQ | 0.667 |
| | CASSLEIFGGIADTDTQYF | LDPVLAELM | LSDPARLTAEL | 0.700 |
| Mycobacterium | CAYRSVPKVSGSRLTF | ANAIPGPRVVTD | AIGRTLVVTD | 0.727 |
| tuberculosis | CSVSDLGVGQPQHF | RLVLLGRLAAQAERLA | RQLLKRLAAEL | 0.667 |
| | | AMPQFLDPGVVFA | ADPQLPVAVA | 0.696 |
| Human immunodeficiency virus 1 | CASSLGQTYEQYF | REMDLAVV | RELIKAVQGV | 0.667 |

# 6 LIMITATIONS

Our study has several limitations. (1) Minimal fuzzy retrieval. The proteome matcher uses position-wise identity for equal-length substrings (and a simple *difflib* ratio for length mismatch). This ignores biochemical similarity and gapped alignments, and can yield both false positives (spurious identity on short windows) and false negatives (near-miss variants). While results shows robustness across thresholds and candidate counts, improved similarity (e.g., substitution-aware scoring or learned aligners) is a clear next step. (2) Confidence as a proxy. The scalar confidence head is not directly supervised. As a result, selective prediction depends on post-hoc thresholds; we plan to add explicit calibration (temperature scaling) and distribution-free coverage control (conformal prediction). (3) Out-of-distribution (OOD) generalization. Performance degrades for unseen pathogens or allele backgrounds; Caution is advised when the data distribution shifts. (4) Data bias. Available corpora are imbalanced across pathogen taxa, allele types, and data sources. Despite cross-validation and reporting with uncertainty, residual bias can influence metrics and qualitative behavior. (5) Scope of signals. The agent operates on tokenized sequences and a string-based matcher; it does not model upstream antigen processing, co-presentation context, or multi-omics signals, which could be important in certain settings.

# 7 ETHICS STATEMENT

All training and evaluation data are publicly available, de-identified, and used under their original licenses. We document sources and licenses in the Appendix and redistribute no raw data. The system is designed for research and for adjunct screening when gold-standard assays are unavailable, infeasible, or inconclusive, and only under expert oversight. It is not a replacement for standard-of-care workflows, and decisions must incorporate domain expertise and orthogonal laboratory evidence. Sequence generators may be misused for designing harmful peptides or probing immune evasion. To mitigate dual-use risks we: (i) release training and inference code; (ii) withhold or provide controlled access to generation weights; and (iii) We do not ship convenience tools for high-throughput peptide exploration, and we encourage community feedback on additional safeguards.

# 8 CONCLUSION

We introduce a new problem framing and a lightweight generate–then–retrieve agent that closes the loop from conditional sequence generation to pathogen-side evidence via a minimal fuzzy matcher. The architecture—shared encoder, staged training, and two causal decoders—delivers consistent gains across tasks, supports selective prediction with abstention, and produces end-to-end reports that aid human interpretation. Experiments demonstrate improvements over strong baselines and practical efficiency.

Looking ahead, we will (i) strengthen retrieval with substitution-aware or learned similarity while preserving the same $\mathcal{R}$ interface, (ii) add explicit calibration and distribution-free coverage control, (iii) expand datasets and bias audits, and (iv) harden release practices for safe access to generative components. We hope this agent perspective catalyzes further work on reliable, interpretable, and deployable host-side diagnostics.

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

APPENDIX

## LLM USABLE STATEMENT

We acknowledge the use of large language models, specifically OpenAI GPT-5, to improve the clarity, grammar, and stylistic consistency of the manuscript, and to help standardize mathematical notation and LaTeX equation formatting. We also used text-to-image generative models to draft and refine schematic illustrations; all figures were curated and finalized by the authors. AI tools were not used for data collection, analysis, experiment design, or for generating scientific claims. The authors independently verified all outputs and take full responsibility for any remaining errors. Only non-sensitive manuscript text and high-level figure descriptions were provided to these tools.

## REPRODUCIBILITY STATEMENT

We provide the complete implementation, pretrained weights, processed datasets, and the exact results used in the paper as supplementary materials attached to the appendix. The package includes training and evaluation scripts, environment specifications, and fixed random seeds so that the main tables and figures can be reproduced within the reported confidence intervals. For safety reasons, a small subset of artifacts that could facilitate misuse—such as long-form, unfiltered sequence generations, fine-grained pathogen annotations, and full-resolution reference repertoires—has been partially redacted or placed under access control. Minor numerical variations may arise from hardware-dependent nondeterminism but remain within the stated confidence bounds.

## A FULL CLINICAL REPORT

This appendix reproduces the complete pathogen detection report for the 60-sample.

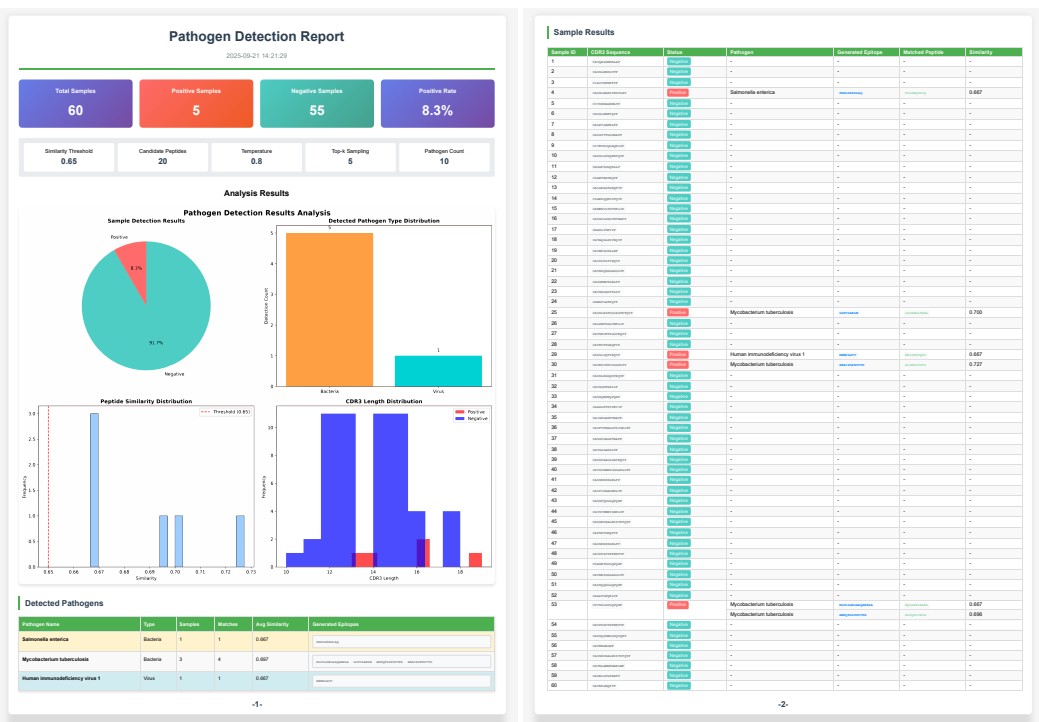

Figure 3: Complete pathogen detection report includes per-sample CDR3, detection status, predicted pathogen, generated epitopes, matched peptides, and similarity scores.

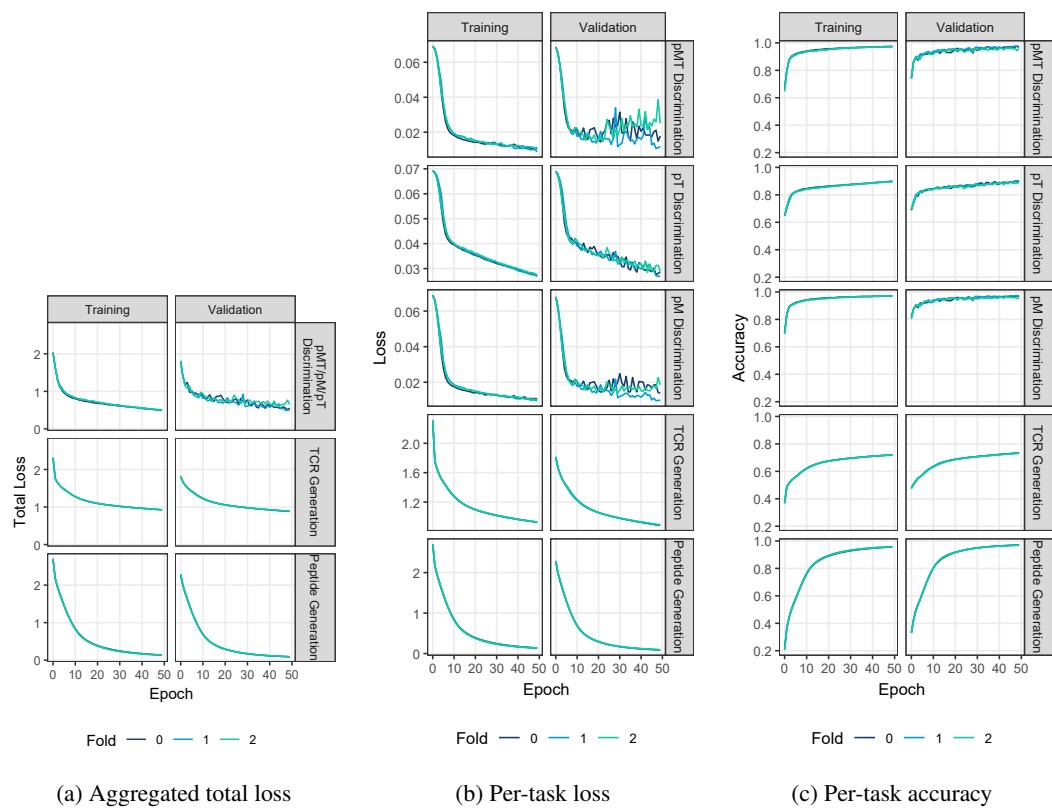

(a) Aggregated total loss      (b) Per-task loss      (c) Per-task accuracy

Figure 4: Training/validation trajectories across tasks. (a) Aggregated total loss. (b) Per-task loss for pMT, pT, pM discrimination, TCR generation, and peptide generation. (c) Per-task accuracy for the same five subtasks. Curves overlay three folds (0/1/2); lower is better for loss and higher is better for accuracy.

## B  TRAINING AND VALIDATION CURVES

This appendix compiles convergence and generalization trajectories for the multi-task setup. Unless noted otherwise, the x-axis is epoch (0–50) and the y-axis is the corresponding metric (loss or accuracy). Each panel overlays training and validation curves and shows three representative folds (Fold 0/1/2). Discrimination tasks include pMT, pT, and pM; generation tasks include TCR generation and peptide generation.

**Total Loss** Figure 4a presents the aggregated total loss across tasks during training and validation, providing a coarse view of optimization speed and potential early-stopping points. Differences across folds reflect variability due to data splits; the train–validation gap offers a quick diagnostic of under/overfitting.

**Per-task Loss** Figure 4b reports training and validation losses for each subtask, enabling a finer-grained view of optimization difficulty and stability. Note that loss scales differ across panels; interpret each panel independently (lower is better).

**Per-task Accuracy** Figure 4c shows accuracy trajectories (0–1, higher is better) for all five subtasks under the same training run. For discrimination, accuracy summarizes pairwise classification performance; for generation, accuracy reflects stepwise agreement with target sequences, aligning with the main-text generation metrics.

# C  Additional details on data resources

**Data resources.** Public repositories such as IEDB, VDJdb, and McPAS-TCR have standardized evaluation but can induce closed-world biases (Vita et al., 2025; Goncharov et al., 2022; Yue et al., 2025). We curate out-of-training-set, clinical-style repertoires and assess both classification (accuracy, AUROC, F1, MCC) and generation (token/sequence accuracy, perfect-ratio) metrics, with agent-level endpoints aligning evaluation with open-world use.

## C.1  Curation pipeline and inclusion criteria

We adopt a repository-agnostic and reproducible pipeline to assemble clinical-style repertoires that better reflect open-world deployment. Records are parsed from public resources, sequence fields are normalized by converting to uppercase and removing gaps and whitespace, and allele nomenclature is standardized to four-digit resolution where available; taxonomy and pathogen labels are harmonized across sources. We then canonicalize sequences by retaining TCR CDR3 amino acid strings, filtering to plausible length ranges, and removing ambiguous residues when they would confound label resolution; V and J gene calls are optionally retained as metadata but are not required inputs. Labels are aligned by standardizing epitope and peptide names, pathogen categories, and assay-derived annotations, and by converting multi-label entries into either one-vs-rest or multi-class targets as required by each task. To control redundancy we perform exact de-duplication within and across sources using a key composed of CDR3aa, peptide, allele, pathogen, and assay, while preserving provenance and clonotype counts as metadata. Split governance follows patient and cohort boundaries with strict disjointness rules to reduce leakage (see Section C.4). Each filtering step writes a manifest in CSV or Parquet that records kept and removed rows together with the reason, enabling deterministic regeneration.

## C.2  Task mapping

We map curated records to the tasks used in this study. The pT discrimination task evaluates TCR–peptide binding or recognition, the pM discrimination task evaluates peptide–MHC binding or presentation, and the pMT discrimination task evaluates the full TCR–peptide–MHC triad. For generation, we consider token-level TCR generation and token-level peptide generation; when sequence-level metrics are defined, we also report sequence accuracy and the fraction of perfect sequences.

## C.3  Split policy and evaluation protocol

Unless stated otherwise, we use K-fold evaluation with fixed random seeds and folds that are disjoint at the patient or cohort level to approximate prospective deployment. Class balance within each fold is preserved by stratifying on task labels and, when available, on HLA allele families. For generation tasks we further enforce target disjointness at the sequence level when feasible, for example epitope-level disjointness across folds for peptide generation. We also report open-world endpoints at the agent level, including peptide reconstruction via Top-k sequence retrieval and pathogen localization via Top-k category retrieval, together with calibrated uncertainty diagnostics.

## C.4  Leakage prevention and quality control

To mitigate closed-world artifacts and data leakage, we apply several safeguards. We perform exact and fuzzy de-duplication across training, validation, and test splits using keys derived from CDR3aa, peptide, allele, and pathogen. We prevent the same peptide–allele pair from appearing in both training and test when such overlap could reduce the task to allele memorization. We optionally enforce cluster-disjoint splits for CDR3 sequences based on sequence-identity thresholds to test generalization beyond near-duplicate sequences. We construct hard negatives matched in length and composition and, where appropriate, sample in-distribution negatives within the same allele or pathogen family to probe specificity. Finally, we validate metadata fields such as allele formats, species tags, and assay types, and remove entries with unresolved ambiguities that could affect labels.

## C.5 Implementation details

Our system is built in PyTorch and follows a shared-encoder, multi-head design that supports both discrimination and generation. Inputs are represented with a learned token embedding and a learned positional embedding; the two are summed and regularized by dropout and layer normalization before entering a stack of Transformer blocks. Each block uses multi-head self-attention with batch-first semantics, a two-layer feed-forward subnetwork with GELU activations and dropout, and residual connections wrapped by layer normalization. Unless otherwise noted, the hidden width is 512, the encoder comprises six layers with eight attention heads, and dropout is set to 0.1.

All classification tasks share the same encoder. A sequence representation is produced by mask-aware averaging over the time dimension to avoid bias from padding, followed by a small bottleneck projection with a hyperbolic tangent nonlinearity. On top of this representation, a linear classifier predicts the binary decision for each task, and a parallel scalar head produces a calibrated confidence score through a sigmoid mapping. In addition, an auxiliary discriminator receives the same pooled representation and outputs the probability that a presented triad is consistent, acting as a lightweight adversarial regularizer during the first training stage.

Sequence generation is handled by two decoder-only modules, one for TCR and one for peptide sequences. Each uses the same embedding scheme as the encoder and a stack of Transformer layers operating under a standard causal mask, without cross-attention. Training uses teacher forcing with a next-token objective; losses ignore padding tokens so that effective batch contributions depend only on valid positions. During inference, decoding proceeds autoregressively with temperature scaling and optional top-$k$ and nucleus sampling; generation halts early when the end-of-sequence token is produced, and completed sequences are masked to prevent spurious continuation.

Training proceeds in three stages with selective freezing. The first stage optimizes the shared encoder together with all classification heads and the discriminator, while both generators are frozen. The loss at this stage is the sum of cross-entropy terms for the three classification tasks and a small adversarial term from the discriminator, scaled to a tenth of the classification magnitude to avoid overpowering the main objective. The second stage focuses on the TCR generator, freezing the classifier heads, the discriminator, and the peptide generator; the objective is the token-level cross-entropy computed with a single right shift. The third stage mirrors the second but trains the peptide generator. This curriculum keeps the encoder anchored by supervised signals before exposing it to the generative objectives.

Optimization uses AdamW with weight decay of 0.01 and a cosine annealing schedule over the total number of update steps in each stage. Gradients are accumulated over four micro-batches to emulate a larger effective batch without exceeding device memory, and global-norm clipping at 1.0 is applied when synchronizing gradients. Mixed-precision training with bfloat16 is enabled, and distributed data parallelism is managed through the Accelerate library, configured to tolerate occasional unused parameters that arise from stage-wise freezing. Metrics are logged throughout training and validation, and the best checkpoint for each stage and fold is selected by the lowest validation loss.

Data are tokenized with a fixed vocabulary and padded to a maximum length of 120 tokens. Attention masks ensure that padding does not contribute to either pooling or attention. We construct three-fold cross-validation splits, maintaining class balance where possible, and report metrics separately for training and validation. For classification, we compute accuracy, precision, recall, F1, ROC–AUC, and the average confidence score over a batch. For generation, we report token accuracy and perplexity derived from the cross-entropy objective, masking out padding to obtain faithful estimates. At inference time, both generators accept an initial context and return only the newly produced continuation so that downstream evaluation can focus on the novel segment.

# D   POLICY CHECKLIST

Table 5: Data-governance policies applied per task. Checkmarks indicate default enforcement; $\sim$ denotes conditional (when metadata are available).

| Policy | pT | pM | pMT | TCR-gen | Pep-gen | Notes |
|---|---|---|---|---|---|---|
| Patient/cohort-disjoint splits | ✓ | ✓ | ✓ | ✓ | ✓ | default K-fold |
| Allele-disjointing for leakage | $\sim$ | ✓ | $\sim$ | | $\sim$ | where feasible |
| (Peptide, allele) pair disjoint | $\sim$ | ✓ | $\sim$ | | ✓ | prevents memorization |
| Target-sequence disjointness | | | | ✓ | ✓ | for generation targets |
| Exact/fuzzy de-dup across splits | ✓ | ✓ | ✓ | ✓ | ✓ | includes near-duplicates |
| Hard-negative sampling | ✓ | ✓ | ✓ | | | length/composition matched |
| HLA allele normalization (4-digit) | $\sim$ | ✓ | ✓ | | | if allele is available |
| Provenance manifest per filter | ✓ | ✓ | ✓ | ✓ | ✓ | deterministic regeneration |

