# OpenReview forum: "An AI Agent for Immune Receptor Fingerprint‑Based Diagnosis of Infection of Unknown Origin"
_ICLR.cc/2026/Conference — ICLR 2026 Conference Withdrawn Submission_

### Official Review · Reviewer_BnLs · 2025-10-27

**Soundness:** 1
**Presentation:** 1
**Contribution:** 1
**Rating:** 2
**Confidence:** 4

**Summary:**

The paper proposes a multi-task sequence-level transformer encoder-decoder approach for predicting pathogenic exposure given the TCR repertoire and HLA typing, using a generative autoregressive likelihood for peptide generation augmented with fuzzy retrieval of proteins from a proteomic database. The encoder is pre-trained with a discriminative objective using labeled data of TCRpMHC triplets. The decoder is trained with an autoregressive likelihood for TCR/peptide generation. Experimental results on classification and generation tasks demonstrate that the proposed approach is competitive relative to baselines.

**Strengths:**

- The paper tackles TCRpMHC, which is an important problem.
- The paper will release code to enhance reproducibility.
- The proposed approach seems competitive with baselines.

**Weaknesses:**

- The writing needs to be improved to enhance clarity:
1)  It's unclear why the paper uses superfluous language to describe their transformer encoder and decoder approach with words such as "AI" and "agents."
- It's unclear how the proposed approach generalizes to unseen peptides or scales, given the overreliance on labeled TCRpMHC datasets:
1)  Scalability: There are an estimated $10^{15}$ TCRs, $10^{5}$ HLAs, and $20^{10}$ peptides. Given that the proposed approach relies on labeled TCR, MHC, and peptide triplets, it will require a significant amount of labeled data to generalize.
2)  Generalization: The paper does not provide experimental results on unseen TCRs/HLAs/peptides. The generation performance seems very low (Table 1). Also, given that the (TCR, HLA) ->  peptide is a many-to-many problem, it's unclear if the proposed setup yields realistic predictions (including Table 4).
- The technical novelty is limited. The proposed approach is a combination of existing approaches with little modification.
- The experimental validation is underwhelming:
1)  Only 60 TCRs are used to validate the approach.
- The paper makes several claims without providing experimental or theoretical evidence:
1)  The paper claims applicability to TCR repertoire data, but evaluations are based on TCRpMHC tasks, and it's unclear how the proposed approach works on TCR repertoire data at the subject-level.
2) The paper claims calibrated uncertainty but does not provide calibration metrics.

**Questions:**

- What is the composition of the TCRpMHC training data in terms of the distribution of TCRs, HLAs, and peptides?
- Table 1: Why is the generation quality for peptide tokens worse than for TCR tokens?
- Given that this is a many-to-many problem, how does the proposed approach account for that?
- How does the proposed approach account for homologous pathogens?

---

### Official Review · Reviewer_LT1u · 2025-10-29

**Soundness:** 3
**Presentation:** 2
**Contribution:** 2
**Rating:** 4
**Confidence:** 3

**Summary:**

This paper addresses the clinically important problem of diagnosing infections of unknown origin (IUO) using only host immune receptor data, rather than direct pathogen detection. It introduces a new machine learning setup for inferring plausible epitopes from a patient's TCR repertoire and HLA type, leveraging this for downstream pathogen localization. The main contributions include a Transformer-based, multi-sequence model pretrained with multiple objectives and a clinically oriented agent employing a generate-then-retrieve paradigm with uncertainty calibration and clinician-in-the-loop adaptation. The model achieves strong performance relative to baselines on several benchmarks, and a complete end-to-end agent with report generation is demonstrated.

**Strengths:**

1. The paper targets a critical and under-explored diagnostic challenge, host-only diagnosis of IUO, which is an appropriate and meaningful new problem for computational immunology and clinical AI.

2. The paper has strong technical foundation. The Transformer-based model benefits from a multi-head, multi-task pretraining regime, sharing encoders for discrimination and conditional generation.

3. The agent explicitly supports abstention under uncertainty and provides actionable confidence scores for both peptide/retrieval outputs. The decision-trace logging underscores the system’s explainability, which is essential for clinical applications.

**Weaknesses:**

1. Despite a substantial literature review, there’s a notable gap in discussion and empirical comparison with very closely related recent transformer-based models for TCR-epitope prediction and generative modeling. For example, methods such as TEIM (X Peng, Y Lei, P Feng, et al. Characterizing the interaction conformation between T-cell receptors and epitopes with deep learning[J]. Nature machine intelligence, 2023, 5(4): 395-407) and TEINet (Y Jiang, M Huo, Li S Cheng. TEINet: a deep learning framework for prediction of TCR–epitope binding specificity[J]. Briefings in bioinformatics, 2023, 24(2): bbad086) are not cited or compared.

2.  Several recent, domain-specific transformer models and conditional generative baselines are missing, especially those which could perform conditional TCR or peptide generation.

3. Although token/sequence accuracy are reported, diversity and expressivity of generated peptides or TCRs are not evaluated. There is no reporting of generation quality beyond accuracy, e.g., diversity, distinct sample counts, or biological plausibility.

4. The sliding-window, identity-based peptide retrieval ignores even basic bioinformatics similarity (e.g., BLOSUM, gapped alignments) and is vulnerable to both false positives and false negatives.

**Questions:**

1. Can the authors justify the exclusion of transformer-based TCR-epitope prediction methods from both Related Work and empirical comparisons?

2. Can you provide more rigorous evaluation of the retrieval module versus basic bioinformatics alignment (e.g., BLOSUM62, Smith-Waterman) on at least a subset of the data?

3. Is there a strategy to enhance generalization for OOD pathogens or rare HLA types?

---

### Official Review · Reviewer_gNs3 · 2025-10-31

**Soundness:** 2
**Presentation:** 1
**Contribution:** 2
**Rating:** 2
**Confidence:** 4

**Summary:**

This paper proposes an AI agent for diagnosing infections from immune receptor data. The model takes T-cell receptor (TCR) sequences and HLA types as priors to generate plausible epitopes, which are then used to retrieve likely pathogen sources, tailored to a patient’s signature. The main contribution lies in combining discriminative and generative training within a multi-stage curriculum. The authors claim state-of-the-art performance on both classification and sequence generation tasks, supported by an extensive (though not always interpretable) experimental study.

**Strengths:**

Creative problem framing, ambitious integration. Connecting and modeling multiple scales of biology is anything but trivial, and this paper presents a solid integration approach.
Generate-then-retrieve loop: elegant design decision in principle. Could constitute a meaningful design decision if it were further substantiated empirically.
Diverse ablations: Multiple architectural ablations are included, showing some experimental thoroughness.

**Weaknesses:**

Underwhelming results
Generation is nearly random: Peptide token accuracy of 0.045 (4.5%) is barely above the 5% random baseline for 20 amino acids. TCR generation at 0.110 (11%) is only marginally better. These numbers appear in Table 1 but are presented as "best" without acknowledging they're objectively inadequate for any real application. The "case demonstration" (Section 5.5) reports 5 positives with similarities of 0.667-0.697 against a threshold of 0.65. These are barely positive, likely false positives, yet are presented as successful detections with no ground truth validation.
Inconsistent and dubiously motivated baselines.
The baselines are heterogeneous and lack conceptual alignment. The paper compares its generative–retrieval architecture against a mix of VAEs (generative), LSTMs (sequence-based), and fully connected ANNs (discriminative), without a consistent objective, input representation, or control for model capacity and supervision level. As a result, the reported improvements are difficult to interpret: they may stem from architectural scale, training regimes, or data preprocessing differences rather than genuine methodological superiority. Furthermore, it is unclear how these baselines relate to the “state-of-the-art” claims, since the tasks themselves are newly defined and lack standardized benchmarks. A principled baseline suite should include models that address the same prediction problem under comparable settings. For example, matched Transformer architectures trained with or without the proposed generative components. Finally, the ablation shows that removing the discriminator or generator shows no performance drops and single task models having comparable performance.
Training procedure and ablations.
The proposed three-stage freeze/unfreeze curriculum (train discriminators → TCR generator → peptide generator) is described in detail but not empirically justified. This staged training may stabilize optimization, but no ablation demonstrates its necessity or advantage over simpler alternatives (e.g., joint or progressively unfrozen training). As it stands, this choice appears heuristic rather than principled. A minimal comparison between (i) the staged schedule, (ii) joint end-to-end training, and (iii) progressive or alternating updates would clarify whether freezing contributes to performance, calibration, or stability.
Conceptual framing (“agentic” contribution).
The “agent” terminology is not clearly justified. The paper repeatedly refers to an “AI agent” or “agentic framework,” but the method as described is a standard feedforward Transformer pipeline with staged training and no genuine perception–planning–action loop, policy optimization, or environment interaction. The use of agentic language adds rhetorical weight without corresponding algorithmic content. If “agent” here merely denotes a multi-objective model with modular components, this should be stated explicitly to avoid overstating the contribution. I recommend removing this terminology or redesigning the system to actually exhibit agentic behavior.
Limited technical novelty.
Beyond its biological framing, the paper primarily reuses existing components. The model architecture, training objectives, and inference mechanism are standard (Transformer encoder–decoder, discriminative and generative heads, string-based retrieval). The “agentic” framing and calibration claims are conceptual embellishments rather than genuine algorithmic innovations. As such, the contribution feels incremental and overstated. The retrieval is primitive: Position-wise character identity on sliding windows (Section 4.4) ignores biochemical similarity, gapped alignments, and PTMs. The authors acknowledge this in limitations but still claim "state-of-the-art performance" throughout.
Clarity and presentation.
This paper is extremely difficult to read. Sentences are frequently overloaded. Both ideas and domains are mixed in single sentences, syntactic separation isn’t immediately clear. A few notable examples in the introduction and preliminary sections:
L42-43 “This host-side signal is scalable to collect, tightly coupled to antigen exposure and naturally suited to machine learning and AI models.”
What is meant by “scalable to collect”? What’s a natural suit for AI models?
L51-52 “A model that, under limited supervision, learns allele specificity and cross-reactivity while producing calibrated uncertainty”
Conflates both biological and statistical modelling, hard to unpack
L210-211 “Indicating interaction feasibility under allele context h, which support auxiliary discriminative objectives.”
Grammatically incorrect, ambiguous (It isn’t immediately clear how interaction feasibility supports the objective).

While use of LLMs is acknowledged in the usability statement, their use does not achieve the intended purpose of improving clarity but instead compounds the issue. I would advocate for extensive stylistic revision of the paper.
Additionally, there’s typing errors scattered throughout: “This work introduce(s)”, “This work develop(s)”. Moreover, the symbol “⊮” appears in the first equation of subsection 4.4 (in the definition of similarity) but is not defined anywhere in the text. To my knowledge, this symbol is not standard in machine learning or bioinformatics notation; if it is meant to denote an indicator or equality function, this should be stated explicitly. Undefined notation makes the equations difficult to interpret.

**Questions:**

What are more appropriate baselines for this task?

What can be improved in the model improve accuracy?

---

### Official Review · Reviewer_X11z · 2025-11-01

**Soundness:** 2
**Presentation:** 2
**Contribution:** 1
**Rating:** 2
**Confidence:** 4

**Summary:**

This paper presents VitalT, an AI system that tries to infer which pathogens a person has been exposed to, using only their immune receptor (TCR) sequences and HLA types without needing pathogen sequencing data.

The model uses a Transformer encoder plus two generative decoders to connect TCRs and peptides, trained under several classification and generation objectives. On top of that, the authors wrap the model inside an agent pipeline that generates candidate peptides, filters them using confidence scores and HLA context, and retrieves likely pathogen sources from a proteome index. Experiments are done on public immune-receptor datasets with custom splits to avoid leakage.

**Strengths:**

Turning immune-receptor analysis into a generation-and-retrieval problem is creative and shows out-of-the-box thinking. The authors clearly describe model stages, optimization steps, and tokenization, which helps reproducibility. They discuss patient-level and allele-disjoint splits, which is a good practice to avoid data leakage.

**Weaknesses:**

1. ANN, CNN, LSTM, and VAE are all older sequence models. The only modern large-scale baseline (ESM-2) is evaluated superficially. Without transformer triad models (e.g., TITAN, TCR-BERT, DeepTCR-Plus), claiming “unique capabilities” seems premature.

2. “Peptide Token Acc.” likely measures next-token prediction, not full-sequence correctness; it’s unclear if padding or masking affects it. No BLEU/perplexity/F1 metrics are reported for generative quality.

3. All models achieve AUC ≥ 0.95, suggesting the dataset may be overly separable or redundant; if the task were biologically realistic, we’d expect lower scores. High uniform AUCs could hint at leakage or easy negatives. AUC is insensitive to class imbalance; no F1, MCC, or calibration metrics are shown, making it impossible to judge practical discriminative quality.

4. VitalT’s AUC gains ( around +0.014 over CNN/LSTM) are numerically small within the 95 % CI overlap. So statistical significance is doubtful. The paper should report paired tests or variance across seeds to confirm this isn’t random noise.

5. It’s unclear whether fine-tuning or frozen ESM-2 was used. Comparing a pretrained protein LM without task-specific tuning may understate its capacity.

6. The sliding-window identity function is ad hoc and not integrated into training, so localization and training objectives are disjoint, which weakens the end-to-end claim. Without supervised calibration or Bayesian interpretation, the scalar confidence seems heuristic; why not use MC-dropout or conformal prediction for uncertainty?

7. While two decoders are trained, the ablation shows minimal improvement. Is this added complexity justified?

8. The 60-sample demo seems cherry-picked; no baseline, no control group, no uncertainty intervals, it’s anecdotal evidence, not validation.

9. Ablation Table 2 shows that removing the generation or discriminator hardly changes classification AUC (differences ≤ 0.002). Improvements come mainly from positional encoding and layer normalization. Each objective contributes meaningfully to the final performance sounds to be overclaimed. The six-objective synergy is not empirically demonstrated.

**Questions:**

1. The “confidence” head is unsupervised.
- How do you know it’s well-calibrated?
- Could you provide calibration plots (ECE, reliability curves) or quantitative evidence of its usefulness?

2. You claim the pathogen index is “dynamically expandable.”
- What happens when the index grows by 100×?
- Have you measured runtime, memory, or accuracy on large proteomes?

3. How do you know that the generated epitopes are biologically meaningful and not just token-level overlaps? Have you tested structural or functional similarity with known complexes?

**Details Of Ethics Concerns:**

The model can generate pathogen-like peptides, raising possible dual-use risks. Claims of clinical relevance are premature since no real patient validation is shown. Clarification and stronger release safeguards are advised.

---

### Note · Authors · 2025-11-16

I have read and agree with the venue's withdrawal policy on behalf of myself and my co-authors.